# Rethinking risk prediction: The role of albumin and other parameters in implant-associated complications after hip or knee arthroplasty

**Petr Domecky**[1], **Anna Rejman Patkova**[1], **Lenka Zaloudkova**[2], **Tomas Kucera**[3], **Pavel Sponer**[3], **Josef Maly**[1]*

1 Department of Social and Clinical Pharmacy, Faculty of Pharmacy in Hradec Kralove, Charles University, Prague, Czech Republic, 2 Department of Clinical Biochemistry and Diagnostics and Osteocenter, University Hospital Hradec Kralove, Hradec Králové, Czech Republic, 3 Department of Orthopedic Surgery, University Hospital Hradec Kralove, Charles University, Faculty of Medicine in Hradec Kralove, Hradec Králové, Czech Republic

* malyj@faf.cuni.cz

**Data Availability Statement:** All relevant data are within the manuscript and its Supporting information files.

## Abstract

### Background

Total hip (THA) or knee (TKA) arthroplasty is still a traumatic and challenging operation that induces inflammation, with a particularly high risk of acute-phase reaction. The aim of this study was to predict the likelihood of implant-associated complications during the preoperative and postoperative course.

### Methods

The prospective observational, non-interventional study of patients diagnosed with primary knee or hip osteoarthrosis undergoing THA or TKA during the study period was conducted. The inflammatory and malnutrition parameters were collected for each patient one day before surgery, two days after surgery, and in outpatient follow-up.

### Results

Of 159 patients analysed, 12 developed implant-associated complications. The albumin, prealbumin, Intensive Care Infection Score (ICIS), Nutritional Risk Index, and white blood cell counts were found to be potential predictors. Notably, preoperative albumin levels significantly differed between groups with and without complications (P-value = 0.042).

### Conclusion

Our study definitively shows that WBC, prealbumin, Nutritional Risk Index, ICIS as a novel marker, and significantly albumin, outperform C-reactive protein in predicting implant-associated complications in hip and knee arthroplasty.

**Funding:** Petr Domecky as a PhD student is supported by Charles University (Project SVV 260 665). The funders had no role in study design, data collection and analysis, decision to publish, or preparation of the manuscript.

**Competing interests:** The authors have declared that no competing interests exist.

## Introduction

Total hip (THA) or knee (TKA) arthroplasty offers pain relief and enhances joint function, stability, and mobility, often in osteoarthritis cases. A condition of the entire joint involving complex pathogenesis with mechanical, inflammatory and metabolic factors. Despite the undoubted benefit and the overall implant survival, arthroplasty is still a traumatic and challenging operation that induces inflammation, especially with an acute phase reaction [1–9]. Various factors can complicate the procedure, and currently, C-reactive protein (CRP) is the generally best known for detecting these implant-associated complications (IAC), highlighting a significant gap in other parameters [10]. The surgery triggers macrophages and monocytes, to release proinflammatory cytokines that drive CRP secretion. CRP levels typically stay elevated for approximately two weeks postoperatively, limiting its predictive capability during recovery. In addition to monitoring these postsurgical consequences, other inflammatory or nutritional parameters are being used to determine the risk of infection or other complications. Neutrophil counts increase due to the cytokines, while lymphocytes are reduced by surgical trauma [1, 8, 11]. Therefore, the neutrophile-lymphocyte (NLR) count could signify underlying complications not immediately evident through CRP levels alone. Furthermore, malnutrition has proven to be a valuable predictor of IAC, therefore, there might be a need for these parameters such as The Prognostic Inflammatory and Nutritional Index (PINI) and Nutritional Risk Index (NRI), which allows for a more comprehensive assessment of patients' status. The Prognostic Inflammatory and Nutritional Index (PINI) was initially used for infection prognosis in critically ill patients, comprising both nutritional and inflammatory parameters [12]. In addition to PINI, the Nutritional Risk Index (NRI) was used to properly assess the influence of malnutrition on the risk of IAC [13]. The Intensive Care Infection Score (ICIS) is a new parameter to characterize the patient's current condition, including infections and their severity [14]. Its application to the orthopaedics field is innovative, aiming to ascertain whether it can offer additional predictive value beyond the capabilities of CRP or the other parameters (NLR, PINI, NLR). Given the limitations of CRP as a standalone parameter to predict the IAC for the abovementioned reasons, we have hypothesized that various malnutrition or inflammatory parameters might offer a superior prediction of IAC during the preoperative and postoperative courses, and these parameters included ICIS and PINI that have not yet been investigated in orthopaedics. If these parameters prove reliable in predicting IAC, it will allow better identification of at-risk patients, leading to better pre- and postoperative care optimisation. A parameter meeting all of the requirements would significantly benefit clinical practice as it could reduce IAC and improve overall patient outcomes.

## Methods

### Study design

This was a prospective observational, non-interventional study of patients diagnosed with primary knee or hip OA undergoing THA or TKA at the Orthopaedics Department of the University Hospital. The recruitment period lasted from 17th May 2020 to 14th July 2022 (extended due to the COVID-19 restrictions).

All THA and TKA performed at our institution were processed in the prospective database, with blood tests performed one day before surgery (preoperative value), two days after surgery and also at planned outpatient follow-up (postoperative value) six to seven weeks after surgery. To investigate the physiological response, the analysis of parameters during outpatient follow-up (postoperative value) was conducted. The diagnostic test accuracy of the proposed method

was assessed when complications occurred during outpatient follow-up and predictive test accuracy of the proposed method was assessed for preoperative and postoperative courses.

## Inclusion and exclusion criteria

The inclusion criteria were age over 18 years, a signed informed consent (written form) with the study protocol and undergoing THA or TKA. Exclusion criteria were clinical signs of infection before surgery, neoplasia, immunosuppression or autoimmune diseases, or ASA (American Society of Anaesthesiologists physical status) score higher than 4.

## Preoperative care

For antibiotic prophylaxis, cefazolin was used; vancomycin was administered in case an allergy to beta-lactams was present. According to the hospital's internal standards, cefazolin should be administered 30 minutes before surgery at a 2–3 g dose according to body weight. The second dose of cefazolin was given 4 hours after the start of surgery and the next two after 6 hours; in total, four doses were administered. Vancomycin was given at 1–1.5 g according to body weight for two doses 12 hours apart. According to the internal hospital standards, nadroparin or enoxaparin was administered as thrombosis prophylaxis, and patients were switched to rivaroxaban after the fourth day.

## Surgical approach and postoperative care

All THAs were performed using the anterolateral surgical approach. All TKAs were performed using the anteromedial surgical approach. From the first postoperative day, rehabilitation, verticalization, walking with crutches training, and limb loading were indicated according to the surgeon's decision. Redon drain was extracted according to the amount of fluid drained on the first or second postoperative day, plus standard analgesic and adequate wound care was administered according to the patient's condition. Sutures were removed on day 10.

## Data collection

CRP, orosomucoid (ORM), ICIS, NLR, prealbumin and albumin samples were collected for each patient at each examination. Blood was separated into aliquots after centrifugation (2,000 x g, 15 minutes). CRP, ORM and prealbumin levels were determined by immunoturbidimetric analysis, whereas albumin was measured by photometric determination using analyser Cobas 8000© (Roche Diagnostics GmbH, Mannheim, Germany). The NLR ratio was calculated based on recorded absolute total neutrophil and lymphocyte counts. ICIS was determined as described in a van der Geest et al. study [14]. PINI was calculated according to the methodology of Vehe et al. [15]. The following formula was used for the NRI: NRI = (14.89 x ALB + 41.7 x (ABW/IBW)), where ALB is albumin, ABW is the actual body weight, and IBW is the ideal body weight. It should be noted that a "higher NRI" suggests an association with desired outcomes. Conversely, a "lower NRI" implies associating lower observed values of NRI with desired outcomes. Therefore, the patient could be at risk for IAC according to lower NRI or higher NRI values [13]. Kellgren-Lawrence classification was used to evaluate osteoarthrosis [16].

The following data were collected from the university hospital's internal database and each patient's medical record: basic demographic and clinical characteristics including osteoarthritis classification and type of prosthesis, and operation characteristic data.

## Study outcomes

The diagnosis of periprosthetic joint infection (PJI) was based on criteria agreed by the International Consensus Meeting group [17]. Furthermore, surgical site infection (SSI) was defined as in the original statement from Mangram et al. [18]. Since only one PJI occurred, the observed outcomes were categorized as IAC (overall complications which may have been related to THA or TKA, including SSI, pulmonary embolism, and infection in sites other than the incision). Based on the absence/presence of IAC or SSI, patients were divided into two groups. The monitored parameters were compared between these groups.

## Statistical analysis

The IBM SPSS version 27 (SPSS Inc., Chicago, IL, USA) was used for the statistical analysis, and a P-value of <0.05 was considered statistically significant. The diagnostic accuracy of the performed test was calculated based on the receiver operating characteristic (ROC) curve function, the area under the curve (AUC), sensitivity, and specificity. When the observed parameters were not determined during outpatient follow-up, these data were still included in the analyses as missing values (i.e. if CRP was not recorded in a patient, the patient was still included in other analyses for different parameters). None of the patients got missing values for all of the observed parameters in outpatient follow-up. Differences between continuous variables were analysed using the t-test for normally distributed data. When the data did not follow a normal distribution, the Mann-Whitney U test was utilised to analyse differences. Furthermore, Fisher's exact test was used for categorical variables. Descriptive statistics (received values, means, standard deviations, and range) were used to present sample characteristics, while categorical variables were presented as frequencies and percentages.

## Ethics approval and consent to participate

This study received approval from the Ethics Committee of the University Hospital Hradec Kralove, ensuring that all research protocols adhered to established ethical guidelines and standards (202001 S08P). All participants were required to sign an informed consent in a written form, which had been approved by the Ethics Committee, as a prerequisite to their participation in the study.

## Results

Over 31 months, 168 patients were included in this prospective observational non-interventional study, of which 159 patients (74 men and 85 women) were included in the final analysis. Nine patients were excluded due to the exclusion criteria (immunosuppression). The average age of all patients was 66.75 ± 9.30 (min. 44 to max. 84) years. The average duration of surgery was 69.78 ± 18.36 (min. 40 to max. 130) minutes. Of all the patients, 102 underwent THA, while 57 underwent TKA. Cemented prosthesis was used in 97 patients, and non-cemented prosthesis in 62 patients. Osteoarthritis grades 3, 3–4, and 4 were identified in 35, 28, and 96 patients, respectively. Twelve patients (7.5%) developed postoperative IAC, including six SSI (3,8%). Other basic characteristics of patients, such as surgery and antibiotic prophylaxis, are presented in Table 1. A comparison between basic characteristics according to the IAC status is presented in Table 2. No differences were observed.

There was a significant difference in albumin value (P-value = 0.042) at the preoperative laboratory test between both groups related to IAC. The remaining observed parameters mean, and their differences between both groups related to IAC and SSI are presented in Table 3. The ROC analysis revealed that only very few parameters had an AUC above 0.6.

**Table 1. Basic characteristics of patients, surgery, antibiotic prophylaxis and complications.**

| Parameter | N = 159 |
|---|---|
| Number of patients; n (%) | 159 (100.00) |
| Average age of patients; mean ± SD | 66,75 ± 9,30 |
| Range of age; min-max | 44–84 |
| Gender ratio (M: F); ratio | 1.14:1 |
| *BMI; n (%)* | |
| Normal (18.5–24.9) | 18 (11.32) |
| Overweight (25–29.9) | 65 (40.88) |
| Obese (30–34.9) | 55 (34.59) |
| Extreme obese (35<) | 21 (13.21) |
| History of COVID-19; n (%) | 24 (15.09) |
| Arterial Hypertension; n (%) | 117 (73.58) |
| Diabetes mellitus; n (%) | 29 (18.23) |
| *ASA; n (%)* | |
| I | 1 (0.63) |
| II | 102 (64.15) |
| III | 55 (34.59) |
| IV | 1 (0.63) |
| *Osteoarthritis classification; n (%)* | |
| 3 | 35 (22.01) |
| 3–4 | 28 (17.61) |
| 4 | 96 (60.38) |
| *Cemented prosthesis; n (%)* | 97 (61.01) |
| Knee* | 52 (53.61) |
| Hip* | 45 (46.39) |
| *Non-cemented prosthesis; n (%)* | 62 (38.99) |
| Knee* | 3 (4.84) |
| Hip* | 59 (95.16) |
| Average length of hospitalization (d); mean ± SD | 11.00 ± 4.50 |
| Range hospitalization length (d); min-max | 7–11 |
| Average length of surgery (minutes); mean ± SD | 69,78 + 18,36 |
| Length of surgery (minutes); min-max | 40–130 |
| *Antibiotic prophylaxis; n (%)* | |
| Allergy to beta-lactams ATB | 29 (18.24) |
| Cefazolin | 131 (82.39) |
| Vankomycin | 28 (17.61) |
| *Implant-associated complications; n (%)* | 12 (7.55) |
| Superficial infection | 5 (3.14) |
| Periprosthetic joint infection | 1 (0.63) |
| Thromboembolism | 3 (1.89) |
| Mitigated urinary tract infection | 2 (1.26) |
| Peroneal nerve paralysis | 1 (0.63) |

ASA—American Society of Anaesthesiologists physical status; ATB—antibiotic; BMI—body mass index; d—days; F—female; M—male; max—maximum; min—minimum; N—denominator; n—absolute value; SD—standard deviation.

* the relative numbers are according to the sum of patients in the category

**Table 2. Comparison of population characteristics between patients according to IAC status.**

| Parameter | IAC-negative N = 147 | IAC-positive N = 12 | P-value |
|---|---|---|---|
| Average age of patients; mean ± SD | 66.70 ± 9.2 | 67.33 ± 10.4 | 0.590 |
| Male; n (%) | 68 (46.26) | 6 (50.00) | 0.518 |
| BMI; n (%) | | | |
| Normal (18.5–24.9) | 17 (11.56) | 1 (8.33) | 0.884 |
| Overweight (25–29.9) | 61 (41.50) | 4 (33.33) | |
| Obese (30–34.9) | 50 (34.01) | 5 (41.67) | |
| Extreme obese (35<) | 19 (12.93) | 2 (16.67) | |
| History of COVID-19; n (%) | 21 (14.29) | 3 (25.00) | 0.393 |
| Arterial Hypertension; n (%) | 109 (74.15) | 8 (66.67) | 0.518 |
| Diabetes mellitus; n (%) | 26 (17.69) | 3 (25.00) | 0.709 |
| ASA; n (%) | | | |
| I | 1 (0.68) | 0 (0.00) | 0.331 |
| II | 92 (62.59) | 10 (83.33) | |
| III | 53 (36.05) | 2 (16.67) | |
| IV | 1 (0.68) | 0 (0.00) | |
| Osteoarthritis classification; n (%) | | | |
| 3 | 32 (21.77) | 3 (25.00) | 0.228 |
| 3–4 | 24 (16.33) | 4 (33.33) | |
| 4 | 91 (61.90) | 5 (41.67) | |
| Cemented prosthesis; n (%) | 89 (60.54) | 8 (66.67) | |
| Knee* | 47 (52.81) | 5 (62.50) | 0.721 |
| Hip* | 42 (47.19) | 3 (37.50) | |
| Non-cemented prosthesis; n (%) | 58 (39.46) | 4 (33.33) | |
| Knee* | 3 (5.17) | 0 (0.00) | 0.641 |
| Hip* | 55 (94.83) | 4 (100.00) | |
| Average length of surgery (minutes); mean ± SD | 69.55 ± 18.4 | 72.58 ± 18.3 | 0.481 |
| Antibiotic prophylaxis; n (%) | | | |
| Cefazolin | 122 (82.99) | 9 (75.00) | 0.444 |
| Vankomycin | 25 (17.01) | 3 (25.00) | |

ASA—American Society of Anaesthesiologists physical status; BMI—body mass index; IAC—implant-associated complication; N—denominator; SD—standard deviation;

* the relative numbers are according to the sum of patients in the category

Specifically, albumin levels lower than 43.0 g/l prior to surgery and 45.1 g/l in the outpatient follow-up were the best cut-offs for the association with IAC and albumin levels lower than 41.1 g/l prior to surgery, and 45.1 g/l in outpatient follow-up were the best cut-offs for the association with SSI. Values higher than 1.5 and 0.5 were the best cut-off values for the association between SSI and ICIS two days after surgery and in outpatient follow-up, respectively. Regarding prealbumin, it was observed that values lower than 0.25 g/l during the outpatient follow-up surgery were significantly associated with the presence of SSI. Furthermore, values higher than 110.5 were the best cut-offs for the association between IAC and NRI two days after surgery. Finally, regarding the WBC, values higher than $6.93 \times 10^9$/L prior to surgery were the best cut-offs for association with IAC. Other values with cut-offs, AUC, 95% confidence interval (CI), sensitivity and specificity are presented in Table 4.

**Table 3. Laboratory findings related to IAC and SSI.**

| Variables | IAC-positive N = 12 | IAC-negative N = 147 | P-value | Variables | SSI-positive N = 6 | SSI-negative N = 153 | P-value |
|---|---|---|---|---|---|---|---|
| **PRE** | **mean ± SD** | **mean ± SD** | | **PRE** | **mean ± SD** | **mean ± SD** | |
| CRP; mg/L | 2.9 ± 2.6 | 2.8 ± 4.1 | 0.611 | CRP; mg/L | 2.2 ± 2.1 | 2.8 ± 4.1 | 0.990 |
| WBC; × $10^9$/L | 7.44 ± 1.23 | 6.68 ± 1.58 | 0.053 | WBC; × $10^9$/L | 7.14 ± 1.41 | 6.73 ± 1.57 | 0.385 |
| NLR | 2.45 ± 1.00 | 2.42 ± 1.01 | 0.772 | NLR | 2.57 ± 1.06 | 2.41 ± 1.00 | 0.343 |
| ICIS | 0.8 ± 0.8 | 0.6 ± 0.8 | 0.319 | ICIS | 0.9 ± 0.9 | 0.6 ± 0.8 | 0.433 |
| ORM; g/L | 0.73 ± 0.23 | 0.76 ± 0.19 | 0.723 | ORM; g/L | 0.68 ± 0.20 | 0.76 ± 0.20 | 0.480 |
| PREALBUMIN; g/L | 0.25 ± 0.04 | 0.26 ± 0.05 | 0.914 | PREALBUMIN; g/L | 0.25 ± 0.03 | 0.26 ± 0.05 | 0.363 |
| ALBUMIN; g/L | 39.6 ± 12.5 | 43.3 ± 6.5 | **0.042** | ALBUMIN; g/L | 30.6 ± 20.9 | 43.6 ± 5.4 | 0.076 |
| PINI | 1.4 ± 4.2 | 0.3 ± 0.8 | 0.187 | PINI | 2.2 ± 5.5 | 0.3 ± 0.8 | 0.195 |
| NRI | 120.8 ± 11.6 | 119.0 ± 23.0 | 0.318 | NRI | 114.5 ± 29.3 | 120.9 ± 11.6 | 0.292 |
| **TWO** | **mean ± SD** | **mean ± SD** | | **TWO** | **mean ± SD** | **mean ± SD** | |
| CRP; mg/L | 133.2 ± 38.5 | 128.3 ± 45.8 | 0.586 | CRP; mg/L | 126.8 ± 44.2 | 128.8 ± 45.3 | 0.943 |
| WBC; × $10^9$/L | 8.51 ± 1.96 | 8.11 ± 2.20 | 0.373 | WBC; × $10^9$/L | 8.50 ± 1.95 | 8.12 ± 2.20 | 0.401 |
| NLR | 4.18 ± 1.24 | 4.12 ± 1.78 | 0.754 | NLR | 4.46 ± 1.53 | 4.12 ± 1.76 | 0.518 |
| ICIS | 1.6 ± 0.9 | 1.5 ± 1.2 | 0.439 | ICIS | 1.9 ± 0.9 | 1.5 ± 1.2 | 0.332 |
| ORM; g/L | 1.13 ± 0.24 | 1.12 ± 0.21 | 0.577 | ORM; g/L | 1.08 ± 0.28 | 1.12 ± 0.21 | 0.892 |
| PREALBUMIN; g/L | 0.14 ± 0.03 | 0.15 ± 0.05 | 0.351 | PREALBUMIN; g/L | 0.13 ± 0.03 | 0.15 ± 0.05 | 0.494 |
| ALBUMIN; g/L | 34.3 ± 4.6 | 33.4 ± 3.3 | 0.197 | ALBUMIN; g/L | 31.8 ± 4.5 | 33.5 ± 3.3 | 0.442 |
| PINI | 35.0 ± 22.3 | 32.4 ± 18.9 | 0.321 | PINI | 37.1 ± 30.1 | 32.4 ± 18.6 | 0.902 |
| NRI | 111.1 ± 8.6 | 105.7 ± 10.5 | 0.073 | NRI | 108.9 ± 7.3 | 106.0 ± 10.6 | 0.455 |
| **POST** | **mean ± SD** | **mean ± SD** | | **POST** | **mean ± SD** | **mean ± SD** | |
| CRP; mg/L | 4.6 ± 3.7 | 4.7 ± 7.4 | 0.567 | CRP; mg/L | 7.4 ± 8.9 | 4.6 ± 7.1 | 0.281 |
| WBC; × $10^9$/L | 7.28 ± 1.20 | 7.05 ± 1.62 | 0.400 | WBC; × $10^9$/L | 6.87 ± 1.22 | 7.08 ± 1.60 | 0.781 |
| NLR | 2.00 ± 0.80 | 2.06 ± 0.70 | 0.857 | NLR | 2.03 ± 0.60 | 2.06 ± 0.71 | 0.795 |
| ICIS | 0.7 ± 0.9 | 0.7 ± 0.9 | 0.470 | ICIS | 1.1 ± 1.2 | 0.7 ± 0.9 | 0.087 |
| ORM; g/L | 0.89 ± 0.27 | 0.97 ± 0.26 | 0.544 | ORM; g/L | 0.97 ± 0.40 | 0.97 ± 0.25 | 0.768 |
| PREALBUMIN; g/L | 0.26 ± 0.04 | 0.45 ± 1.65 | 0.839 | PREALBUMIN; g/L | 0.23 ± 0.05 | 0.44 ± 1.62 | 0.121 |
| ALBUMIN; g/L | 44.5 ± 2.9 | 45.4 ± 3.0 | 0.304 | ALBUMIN; g/L | 42.7 ± 5.0 | 45.5 ± 2.8 | 0.099 |
| PINI | 0.4 ± 0.4 | 0.6 ± 1.8 | 0.561 | PINI | 1.6 ± 3.2 | 0.6 ±1.7 | 0.232 |
| NRI | 126.3 ± 8.8 | 123.5 ± 9.0 | 0.437 | NRI | 127.2 ± 10.4 | 123.6 ± 9.0 | 0.540 |

CRP—c-reactive protein; IAC—implant-associated complication; ICIS—the Intensive Care Infection Score; N—denominator; NLR—neutrophil to lymphocyte ratio; NRI—nutritional risk index; ORM—orosomucoid; PRE—preoperative value; PINI—the Prognostic Inflammatory and Nutritional Index; POST—postoperative value (6—7 weeks after surgery); PRE—preoperative value; SD—standard deviation; SSI—surgical site infection; TWO—two days after surgery value; WBC—white blood cells.

## Discussion

The primary finding of our study was a significant difference in preoperative albumin levels between patients who developed IAC postoperatively and those who did not. Despite the lack of statistically significant difference between the group with and without IAC in other observed parameters, the ROC analysis revealed value insights. Albumin, WBC, and NRI are better related than CRP to IAC, and albumin, prealbumin, and ICIS are better associated with SSI. This is the first study to suggest a wide range of parameters as potential predictors of IAC and SSI. Furthermore, this is the first time ICIS and PINI have been analysed as parameters for predicting IAC or SSI. This approach could provide more personalised risk stratification to prevent negative outcomes after THA or TKA.

**Table 4. Evaluation of diagnostic test accuracy.**

| Time | AUC | CI | Cut-off | Sensitivity | Specificity |
|---|---|---|---|---|---|
| **ALBUMIN-IAC** | | | | | |
| PRE | **0.628** | 0.478–0.777 | 43.0 | 69.20% | 61.40% |
| TWO | 0.424 | 0.252–0.597 | 33.7 | 53.80% | 32.40% |
| POST | **0.625** | 0.456–0.794 | 45.1 | 69.20% | 55.20% |
| **ALBUMIN-SSI** | | | | | |
| PRE | **0.698** | 0.480–0.916 | 41.1 | 71.40% | 74.80% |
| TWO | 0.586 | 0.349–0.824 | 33.7 | 71.40% | 44.40% |
| POST | **0.685** | 0.459–0.910 | 45.1 | 85.70% | 55.00% |
| **CRP-IAC** | | | | | |
| PRE | 0.540 | 0.367–0.713 | 1.4 | 69.20% | 45.50% |
| TWO | 0.520 | 0.368–0.672 | 123.6 | 61.50% | 49.70% |
| POST | 0.554 | 0.367–0.740 | 3.9 | 58.30% | 65.70% |
| **CRP-SSI** | | | | | |
| PRE | 0.501 | 0.302–0.701 | 1.4 | 71.40% | 48.30% |
| TWO | 0.492 | 0.270–0.714 | 139.1 | 57.10% | 61.60% |
| POST | 0.559 | 0.319–0.799 | 3.9 | 66.70% | 65.10% |
| **ICIS-IAC** | | | | | |
| PRE | 0.536 | 0.377–0.694 | 0.5 | 53.80% | 55.50% |
| TWO | 0.523 | 0.385–0.661 | 1.5 | 46.20% | 53.80% |
| POST | 0.538 | 0.364–0.711 | 0.5 | 46.20% | 57.30% |
| **ICIS-SSI** | | | | | |
| PRE | 0.579 | 0.353–0.805 | 0.5 | 57.10% | 55.30% |
| TWO | **0.605** | 0.432–0.778 | 1.5 | 57.10% | 54.30% |
| POST | **0.613** | 0.377–0.850 | 0.5 | 57.10% | 57.70% |
| **NLR-IAC** | | | | | |
| PRE | 0.527 | 0.385–0.669 | 2.34 | 53.80% | 54.10% |
| TWO | 0.517 | 0.388–0.646 | 3.95 | 53.80% | 50.70% |
| POST | 0.484 | 0.294–0.675 | 2.04 | 58.30% | 52.40% |
| **NLR-SSI** | | | | | |
| PRE | 0.544 | 0.381–0.707 | 2.18 | 57.10% | 47.40% |
| TWO | 0.572 | 0.372–0.772 | 3.95 | 57.10% | 50.70% |
| POST | 0.531 | 0.309–0.754 | 2.21 | 66.70% | 64.40% |
| **LOWER-NRI-SSI** | | | | | |
| PRE | 0.520 | 0.268–0.771 | 121.2 | 57.10% | 51.30% |
| TWO | 0.448 | 0.282–0.615 | 103.8 | 57.10% | 56.60% |
| POST | 0.493 | 0.251–0.736 | 125.0 | 57.10% | 48.00% |
| **LOWER-NRI-IAC** | | | | | |
| PRE | 0.469 | 0.296–0.643 | 124.2 | 53.80% | 41.80% |
| TWO | 0.369 | 0.226–0.511 | 110.6 | 53.80% | 33.60% |
| POST | 0.463 | 0.293–0.633 | 126.6 | 61.50% | 38.40% |
| **HIGHER-NRI-IAC** | | | | | |
| PRE | 0.531 | 0.357–0.704 | 121.0 | 69.20% | 48.60% |
| TWO | **0.631** | 0.489–0.774 | 110.5 | 53.80% | 66.40% |
| POST | 0.568 | 0.399–0.736 | 126.2 | 58.30% | 59.40% |
| **HIGHER-NRI-SSI** | | | | | |
| PRE | 0.480 | 0.229–0.732 | 121.0 | 71.40% | 48.00% |
| TWO | 0.552 | 0.385–0.718 | 102.9 | 71.40% | 42.80% |

*(Continued)*

**Table 4.** (Continued)

| Time | AUC | CI | Cut-off | Sensitivity | Specificity |
|---|---|---|---|---|---|
| POST | 0.507 | 0.264–0.749 | 124.7 | 57.10% | 52.00% |
| **OROSOMUKOID-IAC** | | | | | |
| PRE | 0.476 | 0.304–0.648 | 0.70 | 53.80% | 42.10% |
| TWO | 0.553 | 0.386–0.720 | 1.21 | 53.80% | 67.60% |
| POST | 0.452 | 0.253–0.650 | 0.94 | 58.30% | 51.70% |
| **OROSOMUKOID-SSI** | | | | | |
| PRE | 0.421 | 0.210–0.632 | 0.69 | 71.40% | 39.10% |
| TWO | 0.485 | 0.239–0.730 | 1.17 | 57.10% | 59.60% |
| POST | 0.536 | 0.272–0.799 | 0.94 | 71.40% | 52.00% |
| **PINI-IAC** | | | | | |
| PRE | 0.590 | 0.414–0.765 | 0.1 | 69.20% | 60.00% |
| TWO | 0.511 | 0.361–0.660 | 30.3 | 53.80% | 53.80% |
| POST | 0.554 | 0.366–0.743 | 0.4 | 58.30% | 69.90% |
| **PINI-SSI** | | | | | |
| PRE | 0.578 | 0.345–0.812 | 0.1 | 71.40% | 59.60% |
| TWO | 0.514 | 0.261–0.767 | 34.7 | 57.10% | 64.90% |
| POST | 0.574 | 0.324–0.824 | 0.4 | 66.70% | 69.10% |
| **PREALBUMIN-IAC** | | | | | |
| PRE | 0.536 | 0.380–0.691 | 0.26 | 69.20% | 44.10% |
| TWO | 0.523 | 0.369–0.677 | 0.14 | 53.80% | 54.50% |
| POST | 0.558 | 0.413–0.702 | 0.26 | 61.50% | 49.00% |
| **PREALBUMIN-SSI** | | | | | |
| PRE | 0.527 | 0.348–0.706 | 0.25 | 71.40% | 51.70% |
| TWO | 0.576 | 0.382–0.770 | 0.14 | 57.10% | 54.30% |
| POST | **0.673** | 0.470–0.877 | 0.25 | 71.40% | 61.10% |
| **WBC-IAC** | | | | | |
| PRE | **0.659** | 0.523–0.794 | 6.93 | 69.20% | 60.30% |
| TWO | 0.586 | 0.428–0.744 | 7.76 | 69.20% | 50.70% |
| POST | 0.569 | 0.425–0.712 | 6.83 | 69.20% | 48.30% |
| **WBC-SSI** | | | | | |
| PRE | 0.597 | 0.406–0.789 | 6.53 | 71.40% | 53.30% |
| TWO | 0.594 | 0.364–0.824 | 8.59 | 71.40% | 66.40% |
| POST | 0.469 | 0.276–0.662 | 6.83 | 57.10% | 47.70% |

AUC—area under the curve; CI—confidence interval; CRP—c-reactive protein; IAC—implant-associated complication; ICIS—the Intensive Care Infection Score; NLR—neutrophil to lymphocyte ratio; NRI—nutritional risk index; PRE—preoperative value; PINI—the Prognostic Inflammatory and Nutritional Index; POST—postoperative value (6—7 weeks after surgery); SSI—surgical site infection; TWO—two days after surgery value; WBC—white blood cell.

During our study, we collected data on 159 patients. The mean age of the patients was 66.75 ± 9.30, corresponding to the epidemiology of primary OA [3]. Only 4 (2.52%) patients did not have a control sample taken at the scheduled follow-up. Thus, statistical analysis of nutritional and inflammatory parameters was possibly performed on the low frequency of missing samples.

IAC occurred in 12 (7.55%) patients, 5 (3.14%) patients had superficial infection, and one patient with PJI underwent revision. All superficial infections have healed quickly without consequences for the patient. Mitigated urinary tract infections occurred in two patients (1.26%) and were treated using susceptible antibiotics without a negative impact on

arthroplasty. Three (1.89%) patients suffered from thromboembolism after TKA or THA and were managed according to the internal hospital guidelines. One (0.63%) patient with peroneal nerve paralysis after THA was taken in the care of a physiotherapist and underwent appropriate procedures.

A study by Yombi et al. [19] looked at the kinetics of NLR after TKA. It aimed to compare the kinetics of CRP and NLR parameters. This study excluded patients with inflammatory diseases and patients who developed PJI or other inflammatory conditions. Unfortunately, this study did not use the ROC curve for the diagnostic accuracy test. However, in the study by Zhao et al. [20], patients were divided into two groups: the first group consisted of patients with early PJI (26 patients), while the second group consisted of patients without PJI. Early PJI in this study was considered to have developed within four weeks after surgery. Sampling for testing of diagnostic parameters was carried out in cases when any kind of joint related abnormality was observed. The AUC of the ROC curve for NLR was 0.93, and with a cut-off value of 2.77, the sensitivity and specificity were 84.6% and 89.7%, respectively. These data contrasted with our measurements: NLR two days after surgery—AUC: 0.572 (95% CI: 0.372–0.772), cut-off 3.95, sensitivity 57.10%, specificity 50.70%; outpatient follow-up—AUC: 0.531 (95% CI: 0.309–0.754), cut-off 2.21, sensitivity 66.70%, specificity 64.40%. This difference might be explained by the fact that NLR was analysed in all patients in our study. Another reason could also be that we did not assess only the incidence of PJI but also SSI in general.

Blevins et al. [21] evaluated malnutrition as a risk factor for PJI in a retrospective study. The AUC value of the ROC curve reached 0.610, with an albumin cut-off value of 35 g/l, a sensitivity of 11.6%, and a specificity of 98.6%. The similar result was obtained in meta-analysis by Yuwen et al. [22], which found a 2.5-fold increased risk of SSI for albumin levels < 35 g/l. However, the authors did not statistically evaluate the value of albumin using the ROC method, so their results cannot be compared with our study. In our study the preoperative albumin reached the AUC value of 0.628 (95% CI: 0.478–0.777) for IAC, cut-off 43.0 g/l, sensitivity 69.20%, specificity 61.40%, and the AUC value of 0.698 (95% CI: 0.480–0.916) for SSI, sensitivity 71.40%, and specificity 74.80%. Moreover, the difference between albumin preoperative values was statistically significant between IAC positive and IAC negative groups (P-value = 0.042). This result is consistent with a study by Jiang et al. [23], which confirmed that lower preoperative albumin level is an independent risk factor for acute infection after primary total joint arthroplasty (P = 0.015). This risk increased as preoperative albumin levels decreased. Our findings and findings from the mentioned studies support the need for nutritional screening and management prior to elective primary total joint arthroplasty.

Fang et al. [24] included 191,087 patients undergoing TKA or THA in their retrospective study. The study aimed to examine whether lower NRI values were associated with postoperative complications (within 30 days after surgery). NRI was found to better detect malnutrition than albumin. The odds ratio for developing SSI in patients with NRIs between 92–98 was 1.116, while for NRIs < 92 the odds ratio was 1.855. However, no ROC curve analysis was performed here. In our study, higher NRI values proved better at diagnostic accuracy, which could indicated the overnutrition status.

ICIS performed better in SSI prediction than the reference standard represented by CRP. No other study has investigated the feasibility of using ICIS in this patient population. In our study, the AUC for ICIS two days postoperatively was 0.605 (95% CI: 0.432–0.778), the sensitivity: 57.10%, and the specificity: 54.30%, and while in the outpatient follow-up, the AUC was 0.613 (95% CI: 0.377–0.850), the sensitivity: 57.10%, and the specificity: 57.70%. Nevertheless, it should be noted that the cut-off was 0.5 (one day prior to surgery and in outpatient follow-up) or 1.5 (two days after surgery), and ICIS was set to whole numbers.

Other factors could influence the risk of IAC after TKA or THA, as suggested in the article by Basile et al. [25] Among the malnutrition, which we also included in our study as NRI or albumin/prealbumin, intraoperative factors, such as patient-specific factors (e.g. obesity, systemic infections) and surgical conditions (e.g. prolonged surgery, significant blood loss, alongside adherence to aseptic techniques and postoperative care) should be taken into account as risk factors for IAC. Due to this, we have also compared the baseline characteristics of the group with IAC and the group without IAC, as stated in Table 2. Adherence to the aseptic techniques and postoperative care were no issue, as the university hospital had internal preoperative, perioperative and postoperative standards. Each patient received the same internal standard for preoperative care, and also the same internal standard was used during and after the surgery.

CRP is routinely used to diagnose infections, including SSI or PJI in orthopaedics, but the ability to diagnose early PJI, particularly in the postoperative period, and to predict IAC is severely limited. There is a need for different parameters with much faster normalisation of levels and cheaper and better accuracy. Our study showed that another parameter with better AUC, sensitivity and specificity can be used instead of CRP. Moreover, the albumin proved statistically significant differences between analysed groups of patients.

## Limitations

A limiting factor was the small sample size, which produced low outcomes, limiting the possibility of significant statistical results for the evaluated parameters with enough power to detect statistically significant differences. However, even when there is a statistically significant difference, the result should be interpreted cautiously, especially in clinical practice. The short follow-up of the patients was also a drawback, which made only a tiny proportion of IAC possible to be recorded. Therefore, the results are only for the early-term IAC (6–7 weeks as stated in the methodology section) and not for the long-term IAC. A further unforeseen limitation was the COVID-19 disease pandemic, which significantly extended the study period and made it impossible to recruit patients freely. Due to the increase in patients waiting for surgery, patients with worse joint conditions or poorer health may have been prioritized for surgery, potentially affecting laboratory parameters and influencing the study results to some extent. Nevertheless, we analysed both populations regarding their IAC status, and there were no differences. Therefore, this is not a limitation. Furthermore, a significant increase in sample collections during hospitalization would improve our results, eventually enabling a better understanding of the parameters' kinetics. On average, patients were hospitalized for 11 ± 4.5 days, but only one collection after surgery was performed to determine or calculate the levels of laboratory parameters. Unfortunately, due to some insurmountable conditions, such as a lack of financial means for multiple collections and a deficiency in staff availability (i.e., extra work was needed since sample collections were not routinely performed), we could not perform more sample collections.

## Conclusion

The predictive value of albumin, NRI, prealbumin, WBC and the novel marker ICIS in assessing the risk of IAC and SSI was demonstrated in this study. Notably, albumin levels show significant preoperative differences related to IAC status, suggesting that patients with low preoperative albumin levels might benefit from the proper nutritional support to prevent the negative outcomes. The results highlight the critical role of comprehensive inflammatory and nutritional assessments in managing and predicting IAC and SSI, with significant implications for patient risk assessment. Moreover, when considering the results for CRP, our findings

make the results even more important, as CRP, which is conventionally used and preferred, provides worse results than most of the parameters tested. Incorporating other malnutrition or inflammatory parameters among standard perioperative examinations might provide superior outcomes and allow monitoring of patients at higher risk for negative outcomes. Further research with a larger cohort of patients is warranted to substantiate these observations, especially their clinical significance.

## Supporting information

**S1 File. Dataset.**
(XLSX)

## Acknowledgments

We gratefully acknowledge the cooperation with the staff of the Department of Orthopedic Surgery, and the staff of the Haematology laboratory of the IV. Internal haematology clinic, the staff of the Department of Clinical Biochemistry and Diagnostics and Osteocenter, and the English language proofreader.

## Author Contributions

**Conceptualization:** Petr Domecky, Anna Rejman Patkova, Lenka Zaloudkova, Josef Maly.

**Data curation:** Petr Domecky, Anna Rejman Patkova, Josef Maly.

**Formal analysis:** Petr Domecky, Anna Rejman Patkova, Josef Maly.

**Funding acquisition:** Josef Maly.

**Investigation:** Petr Domecky, Anna Rejman Patkova, Josef Maly.

**Methodology:** Petr Domecky, Anna Rejman Patkova, Lenka Zaloudkova, Tomas Kucera, Pavel Sponer, Josef Maly.

**Project administration:** Petr Domecky, Anna Rejman Patkova, Josef Maly.

**Resources:** Petr Domecky, Anna Rejman Patkova, Josef Maly.

**Supervision:** Josef Maly.

**Writing – original draft:** Petr Domecky, Anna Rejman Patkova.

**Writing – review & editing:** Petr Domecky, Anna Rejman Patkova, Lenka Zaloudkova, Tomas Kucera, Pavel Sponer, Josef Maly.

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
