## [Decision Letter · Decision Letter 0]

17 Mar 2024

PONE-D-24-02675Rethinking risk prediction: The role of albumin and other parameters in implant-associated complications after hip or knee arthroplastyPLOS ONE

Dear Dr. Maly,

Thank you for submitting your manuscript to PLOS ONE. After careful consideration, we feel that it has merit but does not fully meet PLOS ONE’s publication criteria as it currently stands. Therefore, we invite you to submit a revised version of the manuscript that addresses the points raised during the review process.

We look forward to receiving your revised manuscript.

Kind regards,

Gennaro Pipino, Md

Academic Editor

PLOS ONE

Journal Requirements:

"Petr Domecky as a PhD student is supported by Charles University (Project SVV 260 665)."

3. In the online submission form, you indicated that [The data that support the findings of this study are available from University Hospital Hradec Kralove but restrictions apply to the availability of these data, which were used under license for the current study, and so are not publicly available. Data are however available from the authors upon reasonable request and with permission of University Hospital Hradec Kralove.]

4. We note you have included a table to which you do not refer in the text of your manuscript. Please ensure that you refer to Table 1 and 2 in your text; if accepted, production will need this reference to link the reader to the Table.

**Additional Editor Comments:**

Please find the comments of our reviewers attached below. We would be grateful if you could make the corrections suggested by the reviewers when revising your article.  Personally I find the article very interesting, and I believe that with some corrections it can be published.

Reviewers' comments:

Reviewer's Responses to Questions

**Comments to the Author**

1. Is the manuscript technically sound, and do the data support the conclusions?

Reviewer #1: Yes

Reviewer #2: Yes

Reviewer #3: Yes

Reviewer #4: No

2. Has the statistical analysis been performed appropriately and rigorously? 

Reviewer #1: Yes

Reviewer #2: Yes

Reviewer #3: Yes

Reviewer #4: No

3. Have the authors made all data underlying the findings in their manuscript fully available?

Reviewer #1: Yes

Reviewer #2: Yes

Reviewer #3: Yes

Reviewer #4: Yes

4. Is the manuscript presented in an intelligible fashion and written in standard English?

Reviewer #1: Yes

Reviewer #2: Yes

Reviewer #3: Yes

Reviewer #4: Yes

5. Review Comments to the Author

Reviewer #1: the aims and objectives of the study is apt, the methodology including the design, data collection, the outcomes and statistical analysis is good.the ethical approval and consent were duly taken. the result was properly analysed

Reviewer #2: I have been given the opportunity to review your article aimed at predicting complications associated with total hip (THA) and knee (TKA) arthroplasty, highlighting the high risk of post-operative inflammatory reactions. The prospective observational study conducted on patients with primary knee or hip osteoarthritis undergoing THA or TKA was described with absolute clarity, identifying albumin, prealbumin, Intensive Care Infection Score (ICIS), Nutritional Risk Index, and white blood cell count as potential predictors. However, it should be noted that prosthetic infection may also result from the complex concomitant action of multiple factors: bacteria, prostheses, and host weakness. In this regard, I would kindly ask you to briefly elaborate on the possible role of host-related concomitant factors, as it is known that the immune response is able to limit infection. I would like to bring to your attention, solely for illustrative purposes, an interesting study (DOI: 10.1186/S10195-021-00607-6) that may be of specific interest to you and could potentially be considered in your references, if deemed useful. This study highlights that among the intraoperative reasons for the development of infectious complications after joint arthroplasty are a duration of surgery exceeding 180 minutes, significant blood loss (more than 800 ml), blood transfusion, excessive tissue trauma, the presence of nosocomial bacterial strains, and failure to adhere to aseptic and antiseptic rules.

Your study clearly demonstrates that parameters such as albumin, NRI, prealbumin, white blood cells, and ICIS can predict implant-related complications, emphasizing the importance of comprehensive inflammatory and nutritional assessments in managing and predicting complications, with significant implications for patient risk assessment. The text discusses the results of the study and refers to previous studies to provide context and comparisons. However, some points may require clarification. In this regard, it is necessary to clarify whether specific measures were taken to minimize the risk of confounding in data collection and statistical analysis. The text mentions some limitations of the study, such as the small sample size and the short follow-up of patients. However, it may be useful to further explore how these limitations may have influenced the results and conclusions of the study. It would be helpful to discuss more thoroughly the clinical implications of the study results and how they may affect daily clinical practice.

In conclusion, the analysis you have drawn provides a good overview of the study's objectives, methods, results, and conclusions, although some parts could be improved with additional clarification and insights, as mentioned above.

Reviewer #3: The manuscript is well-structured and the research question is clearly defined. However, there are some areas that require improvement or clarification.

Major comments:

The rationale for selecting the specific inflammatory and malnutrition parameters should be better justified in the background. While the authors mention that CRP is the most commonly used parameter, it is unclear why the other parameters were chosen.

The sample size calculation is not provided so if it's possible it's better include it.

The authors could provide more details on the surgical procedures and postoperative care in the methods if it possible. This information is important I think to understand potential confounding factors and assess the generalizability of the study findings.

The authors mention that 12 patients developed implant-associated complications. More details on the management, etiology, of complications and their severity could be provided.

While the study identified several potential predictors of IAC, it is unclear how these findings can be used in clinical practice to improve patient outcomes and how. The authors could discuss a little bit in depth the clinical significance of their findings.

The manuscript would benefit from a second round copyediting to improve the grammar, punctuation, and academic style, nut over all was written in very good English.

Minor comments:

The authors could provide more details on the patient population, including more like comorbidities, and medications if it's possible.

The authors could provide a little more details on the data collection and analysis methods, including how missing data were handled.

The authors could provide more details on the statistical analysis, there are any multiple linear regression analyses that you did analyzing the datas? If no, why you did not performed them?

You cited KL score referring to Kohn et all. in the (16.) i think it's better refer to the original classification Kellgren JH, Lawrence JS. The Epidemiology of Chronic Rheumatism. Atlas of Standard Radiographs. Vol 2. Oxford, UK: Blackwell Scientific; 1963.

why it's important for you the grade of arthrosis, did you consider it as factor for ICA in your analysis, or it was not influent?

Overall, the manuscript presents an interesting study with potential clinical implications. The manuscript would benefit from minor revisions to improve the clarity of the study findings.

Reviewer #4: Thank you for the invitation to review. This is a small prospective study of the predictive values of several biochemical markers in detecting post-operative infections in patients undergoing TKA/ THA. There are a few issues:

1) The hypothesis of the study was not stated and it is unclear how the authors had chosen the potential predictors. This creates confusion when interpreting the results. For example, we do not expect albumin or nutritional status to drastically fluctuate from one day before surgery to two days post surgery; are the authors using this predictor to inform the pre-operative risk of infection in patients? WCC, ICIS, NLR and CRP on the other hand are more commonly used as diagnostic aid post-operatively. Note that CRP also peaks on post-operative day 2-3, and while it is commonly used to mark a trend, it is not a good benchmark for comparison with potential predictors at this particular time point.

2) I do not agree with non-implant associated complications such as thromboembolism, urinary tract infection and peroneal nerve paralysis being categorised under implant-associated complications. They are poorly related to the chosen predictors and would invalidate the results of the study.

3) The authors stated there were twelve patients who developed implant-associated complications, including six SSIs, however this did not correspond to data presented in Table 1.

4) Interpretation of the data must be performed in the context of clearly stated hypothesis / aims. I am unable to provide specific comments without clarification of the study aims. Generally speaking, p-values alone are inadequate in terms of identifying significant results; the size of between-group difference and confidence intervals must also be taken into account. For example, the IAC negative group has a pre-operative albumin of 43.3 versus 39.6 in the IAC positive group with a p-value of 0.042. The size of the difference (less than 4 g/L) is small when compared with the SSI groups, where the values were 43.6 and 30.6 meaning a difference of 13 g/L between the SSI-positive and SSI-negative groups. The p-value was 0.076 which is above 0.05 due to the small sample size (n=6) in the SSI positive group.

5) Please update reference source on page 6 line 138

6. PLOS authors have the option to publish the peer review history of their article (what does this mean?). If published, this will include your full peer review and any attached files.

Reviewer #1: **Yes: **Mustapha Alimi

Reviewer #2: **Yes: **Giuseppe Basile

Reviewer #3: No

Reviewer #4: No

---

## [Author Response · Author response to Decision Letter 0]

1 May 2024

Hradec Králové, 01/05/2024

Dear Editor-in-Chief, dear editors

We gratefully acknowledge the editor and reviewers for their valuable comments on our manuscript entitled “Rethinking risk prediction: The role of albumin and other parameters in implant-associated complications after hip or knee arthroplasty”. We have thoroughly considered all the suggestions and recommendations provided and have incorporated them into our revised version of the manuscript. Reviewers’ commentaries regarding statistical analysis were consulted and approved by a statistician, who also contributed to the relevant responses. Moreover, the tables were revised (i.e. NRI values in Table 3), and the other sections have been condensed to emphasize the key point better. Furthermore, we have included all the requirements stated by the Academic editor:

Requirement 1: Please ensure that your manuscript meets PLOS ONE's style requirements, including those for file naming. The PLOS ONE style templates can be found at https://journals.plos.org/plosone/s/file?id=wjVg/PLOSOne_formatting_sample_main_body.pdf and https://journals.plos.org/plosone/s/file?id=ba62/PLOSOne_formatting_sample_title_authors_affiliations.pdf

Authors´ comment: We have checked the revised manuscript according to PLOS ONE’s style requirements.

Requirement 2: Thank you for stating the following financial disclosure: 

"Petr Domecky as a PhD student is supported by Charles University (Project SVV 260 665)."

 Please state what role the funders took in the study. If the funders had no role, please state: ""The funders had no role in study design, data collection and analysis, decision to publish, or preparation of the manuscript."" If this statement is not correct you must amend it as needed. Please include this amended Role of Funder statement in your cover letter; we will change the online submission form on your behalf.

Authors´ comment: The funders had no role in study design, data collection and analysis, decision to publish, or preparation of the manuscript.

Requirement 3: In the online submission form, you indicated that [The data that support the findings of this study are available from University Hospital Hradec Kralove but restrictions apply to the availability of these data, which were used under license for the current study, and so are not publicly available. Data are however available from the authors upon reasonable request and with permission of University Hospital Hradec Kralove.]

Authors´ comment: After consultation with the department where the data collection took place and after careful consideration by the Ethics Committee, the data in anonymised form, including statistical outputs, are attached as a supplement.

Data Availability: All relevant data are within the manuscript and its Supporting Information files. 

Requirement 4: We note you have included a table to which you do not refer in the text of your manuscript. Please ensure that you refer to Table 1 and 2 in your text; if accepted, production will need this reference to link the reader to the Table.

Authors´ comment: Thank you for this commentary. We have updated all the cross-references.

Requirement 5: Please review your reference list to ensure that it is complete and correct. If you have cited papers that have been retracted, please include the rationale for doing so in the manuscript text, or remove these references and replace them with relevant current references. Any changes to the reference list should be mentioned in the rebuttal letter that accompanies your revised manuscript. If you need to cite a retracted article, indicate the article’s retracted status in the References list and also include a citation and full reference for the retraction notice.

Authors´ comment: We have checked all the references. No retracted articles are included in our manuscript. 

We are highly thankful for the time and effort invested in reviewing our work, and we believe that the comments have significantly improved the quality of our research. 

Kind regards

Josef Maly

Department of Social and Clinical Pharmacy

Faculty of Pharmacy in Hradec Kralove, Charles University

Ak. Heyrovskeho 1203/8, 500 05 Hradec Kralove, Czech Republic

Reviewer: 1

Comments to the Author: The aims and objectives of the study is apt, the methodology including the design, data collection, the outcomes and statistical analysis is good. The ethical approval and consent were duly taken. The result was properly analysed.

Authors´ comment: We greatly appreciated your positive feedback. 

Changes in the text: None.

Reviewer: 2

Comments to the Author: I have been given the opportunity to review your article aimed at predicting complications associated with total hip (THA) and knee (TKA) arthroplasty, highlighting the high risk of post-operative inflammatory reactions. The prospective observational study conducted on patients with primary knee or hip osteoarthritis undergoing THA or TKA was described with absolute clarity, identifying albumin, prealbumin, Intensive Care Infection Score (ICIS), Nutritional Risk Index, and white blood cell count as potential predictors. However, it should be noted that prosthetic infection may also result from the complex concomitant action of multiple factors: bacteria, prostheses, and host weakness. In this regard, I would kindly ask you to briefly elaborate on the possible role of host-related concomitant factors, as it is known that the immune response is able to limit infection. I would like to bring to your attention, solely for illustrative purposes, an interesting study (DOI: 10.1186/S10195-021-00607-6) that may be of specific interest to you and could potentially be considered in your references, if deemed useful. This study highlights that among the intraoperative reasons for the development of infectious complications after joint arthroplasty are a duration of surgery exceeding 180 minutes, significant blood loss (more than 800 ml), blood transfusion, excessive tissue trauma, the presence of nosocomial bacterial strains, and failure to adhere to aseptic and antiseptic rules.

Authors´ comment: Thank you for this suggestion. In the Discussion section, we have elaborated on the possible role of host-related concomitant factors. Furthermore, we have also included the comparison between the possible role of host-related concomitant factors and intraoperative reasons, which were available for our population (see new Table 2).

Changes in the text: A new part added to the discussion section:

Line 255–264: Other factors could influence the risk of IAC after TKA or THA, as suggested in the article by Basile et al. [25] Among the malnutrition, which we also included in our study as NRI or albumin/prealbumin, intraoperative factors, such as patient-specific factors (e.g. obesity, systemic infections) and surgical conditions (e.g. prolonged surgery, significant blood loss, alongside adherence to aseptic techniques and postoperative care) should be taken into account as risk factors for IAC. Due to this, we have also compared the baseline characteristics of the group with IAC and the group without IAC, as stated in Table 2. Adherence to the aseptic techniques and postoperative care were no issue, as the university hospital had internal preoperative, perioperative and postoperative standards. Each patient received the same internal standard for preoperative care, and also the same internal standard was used during and after the surgery.

Page 9: a new Table 2 has been added: Comparison of population characteristics between patients according to IAC status.

Comments to the Author: Your study clearly demonstrates that parameters such as albumin, NRI, prealbumin, white blood cells, and ICIS can predict implant-related complications, emphasizing the importance of comprehensive inflammatory and nutritional assessments in managing and predicting complications, with significant implications for patient risk assessment. The text discusses the results of the study and refers to previous studies to provide context and comparisons. However, some points may require clarification. In this regard, it is necessary to clarify whether specific measures were taken to minimize the risk of confounding in data collection and statistical analysis. The text mentions some limitations of the study, such as the small sample size and the short follow-up of patients. However, it may be useful to further explore how these limitations may have influenced the results and conclusions of the study. It would be helpful to discuss more thoroughly the clinical implications of the study results and how they may affect daily clinical practice. In conclusion, the analysis you have drawn provides a good overview of the study's objectives, methods, results, and conclusions, although some parts could be improved with additional clarification and insights, as mentioned above.

Authors´ comment: Thank you for your comment. We have included information regarding the limitations of the study. About whether specific measures were taken to minimize the risk of confounding in data collection and statistical analysis, we should clarify that we carefully defined inclusion and exclusion criteria to ensure a homogeneous study population even with consecutive samples. Moreover, we could not perform a multivariate analysis (see Reviewer 3 comments), but we compared groups according to the IAC status. No differences were observed (see new Table 2). The conclusion was also edited to include the daily practice recommendations. However, the results might be limited, mainly due to the points in the limitations sections. Furthermore, it should be noted that this response was consulted with a statistician.

Changes in the text: Limitation and conclusion section has been edited:

Line 276–278: Therefore, the results are only for the early-term IAC (6–7 weeks as stated in the methodology section) and not for the long-term IAC.

Line 282–283: Nevertheless, we analysed both populations regarding their IAC status, and there were no differences. Therefore, this is not a limitation.

Line 293–294: Notably, albumin levels show significant preoperative differences related to IAC status and patients with low preoperative albumin levels might benefit from the proper nutritional support to prevent the negative outcomes.

Line 298–302: Incorporating other malnutrition or inflammatory parameters among standard perioperative examinations might provide superior outcomes and allow monitoring of patients at higher risk for negative outcomes. Further research with a larger cohort of patients is warranted to substantiate these observations, especially their clinical significance.

Reviewer: 3

Comments to the Authors:

The rationale for selecting the specific inflammatory and malnutrition parameters should be better justified in the background. While the authors mention that CRP is the most commonly used parameter, it is unclear why the other parameters were chosen.

Authors´ comment: According to the current practice, the CRP is the most commonly used parameter. Regarding its limitations related to its pharmacokinetics, the rationale was to understand how the other parameters work and if they can provide better results. The background has been slightly edited to include the rationale.

Changes in the text: 

Line 57–58: Therefore, the neutrophile-lymphocyte (NLR) count could signify underlying complications not immediately evident through CRP levels alone

Line 58–61: Furthermore, malnutrition has proven to be a valuable predictor of IAC, therefore, there might be a need for these parameters such as The Prognostic Inflammatory and Nutritional Index (PINI) and Nutritional Risk Index (NRI), which allows for a more comprehensive assessment of patients’ status.

Line 66–75: Its application to the orthopaedics field is innovative, aiming to ascertain whether it can offer additional predictive value beyond the capabilities of CRP or the other parameters (NLR, PINI, NLR). Given the limitations of CRP as a standalone parameter to predict the IAC for the abovementioned reasons, we have hypothesized that various malnutrition or inflammatory parameters might offer a superior prediction of IAC during the preoperative and postoperative courses, and these parameters included ICIS and PINI that have not yet been investigated in orthopaedics. If these parameters prove reliable in predicting IAC, it will allow better identification of at-risk patients, leading to better pre- and postoperative care optimisation. A parameter meeting all of the requirements would significantly benefit clinical practice as it could reduce IAC and improve overall patient outcomes.

Comments to the Authors:

The sample size calculation is not provided so if it's possible it's better include it.

Authors´ comment: As this is a pivotal study, the sample size was not predetermined. The initial documentation, which included study design, hypothesis formulation, and study protocol, was prepared in 2018. We operated under the assumption that there might be additional suitable parameters beyond C-reactive protein (CRP) for evaluating IAC. Unfortunately, the pivotal studies examining individual parameters were conducted after our initial preparatory work or did not include analysis of the Receiver Operating Characteristic (ROC) curve with Area Under the Curve (AUC) estimation to precisely define the null hypothesis value (i.e. AUC for null hypothesis):

• Yombi et al. (2016) did not include ROC analysis and focused only on TKA with the neutrophil-to-lymphocyte ratio.

• Zhao et al. (2020) included ROC analysis only for the neutrophil-to-lymphocyte ratio.

• Blevins et al. (2018) provided ROC analysis merely for albumin.

• Yuwen et al. (2017) did not perform ROC analysis and studied only albumin.

• Fang et al. (2022) lacked ROC analysis and focused on the nutritional risk index.

Given the insufficient data on the AUC for various parameters, we could not accurately calculate the sample size for our study. Therefore, we have decided to conduct a consecutive diagnostic accuracy study with a limited period. The primary goal was to publish these findings and conduct a follow-up study on the inflammatory parameters of interest. Unfortunately, our study was impeded by the COVID-19 pandemic, resulting in delayed evaluation of the final results. However, we tried to calculate an approximate sample size based on the expected prevalence of IAC. We assumed the AUC we want to achieve for most parameters (i.e. alternative hypothesis = AUC: 0.6-0.7). The calculation was followed with the settings: Type I error (Alpha, significance): 0.05, type II error (Beta, 1-power): 0.20 and AUC ROC: > 0.6 aimed for the 171 patients and 19 outcomes. This calculation was used only to illustrate how many patients we need in consecutive recruitment. Furthermore, it should be noted that this response was consulted with a statistician.

Changes in the text: None.

Comments to the Authors: The authors could provide more details on the surgical procedures and postoperative care in the methods if it possible. This information is important I think to understand potential confounding factors and assess the generalizability of the study findings.

Authors´ comment: Thank you for this commentary. We have included this part into the methodology section. 

Changes in the text: The new subsection has been added: Preoperative care; Surgical approach and postoperative care.

Line 94–102: Preoperative care

For antibiotic prophylaxis, cefazolin was used; vancomycin was administered in case

---

## [Decision Letter · Decision Letter 1]

18 Jun 2024

Rethinking risk prediction: The role of albumin and other parameters in implant-associated complications after hip or knee arthroplasty

PONE-D-24-02675R1

Dear Dr. Maly,

We’re pleased to inform you that your manuscript has been judged scientifically suitable for publication and will be formally accepted for publication once it meets all outstanding technical requirements.

Kind regards,

Gennaro Pipino, Md

Academic Editor

PLOS ONE

Additional Editor Comments (optional):

All reviewers agree to accept the paper. Congratulations for the excellent work done.

Reviewers' comments:

Reviewer's Responses to Questions

**Comments to the Author**

1. If the authors have adequately addressed your comments raised in a previous round of review and you feel that this manuscript is now acceptable for publication, you may indicate that here to bypass the “Comments to the Author” section, enter your conflict of interest statement in the “Confidential to Editor” section, and submit your "Accept" recommendation.

Reviewer #2: All comments have been addressed

Reviewer #3: All comments have been addressed

Reviewer #4: All comments have been addressed

2. Is the manuscript technically sound, and do the data support the conclusions?

Reviewer #2: Yes

Reviewer #3: Yes

Reviewer #4: Partly

3. Has the statistical analysis been performed appropriately and rigorously? 

Reviewer #2: Yes

Reviewer #3: Yes

Reviewer #4: Yes

4. Have the authors made all data underlying the findings in their manuscript fully available?

Reviewer #2: Yes

Reviewer #3: Yes

Reviewer #4: No

5. Is the manuscript presented in an intelligible fashion and written in standard English?

Reviewer #2: Yes

Reviewer #3: Yes

Reviewer #4: Yes

6. Review Comments to the Author

Reviewer #2: Dear Authors, I have read your additions, finding them entirely consistent and in accordance with my expectations. I consider your work interesting. The purpose is clear and respected. I believe that the information provided is to be considered entirely sufficient and represents useful elements to encourage the development of new scientific work.

Reviewer #3: (No Response)

Reviewer #4: (No Response)

7. PLOS authors have the option to publish the peer review history of their article (what does this mean?). If published, this will include your full peer review and any attached files.

Reviewer #2: **Yes: **Giuseppe Basile

Reviewer #3: No

Reviewer #4: No

---

## [Editor Report · Acceptance letter]

24 Jun 2024

PONE-D-24-02675R1 

PLOS ONE

Dear Dr. Maly, 

I'm pleased to inform you that your manuscript has been deemed suitable for publication in PLOS ONE. Congratulations! Your manuscript is now being handed over to our production team.

Kind regards, 

on behalf of

Professor Gennaro Pipino 

Academic Editor

PLOS ONE